# Tentatively Identified (UPLC/T-TOF–MS/MS) Compounds in the Extract of *Saussurea costus* Roots Exhibit In Vivo Hepatoprotection via Modulation of HNF-1α, Sirtuin-1, C/ebpα, miRNA-34a and miRNA-223

**DOI:** 10.3390/molecules27092802

**Published:** 2022-04-28

**Authors:** Heba A. El Gizawy, Alaadin E. El-Haddad, Amr M. Saadeldeen, Sylvia A. Boshra

**Affiliations:** 1Department of Pharmacognosy, Faculty of Pharmacy, October 6 University (O6U), Giza 12585, Egypt; hebaelgizawy@o6u.edu.eg; 2Department of Pharmacognosy, School of Pharmacy, Newgiza University (NGU), Newgiza, km 22 Cairo-Alexandria Desert Road, Giza 12577, Egypt; amr.saadeldeen@ngu.edu.eg; 3Department of Biochemistry, Faculty of Pharmacy, October 6 University (O6U), Giza 12585, Egypt; sylviaazmy@o6u.edu.eg

**Keywords:** *Saussurea lappa*, Asteraceae, phenolics, hepatoprotective, HNF-1α, C/ebpα, Sirtuin-1, miRNA

## Abstract

*Saussurea costus* is a plant traditionally used for the treatment of several ailments. Our study accomplished the UPLC/T-TOF–MS/MS analysis of a methanol extract of *Saussurea costus* roots (MESC), in addition to lipoidal matter determination and assessment of its in vivo hepatoprotective activity. In this study, we were able to identify the major metabolites in MESC rather than the previously known isolated compounds, improving our knowledge of its chemical constituents. The flavones apigenin, acacetin, baicalein, luteolin, and diosmetin, and the flavonol aglycones quercetin, kaempferol, isorhamnetin, gossypetin, and myricetin and/or their glycosides and glucuronic derivatives were the major identified compounds. The hepatoprotective activity of MESC was evaluated by measuring catalase activity using UV spectrophotometry, inflammatory cytokines and apoptotic markers using ELISA techniques, and genetic markers using PCR. Paracetamol toxicity caused a significant increase in plasma caspase 2, cytokeratin 18 (CK18), liver tumor necrosis factor-α (TNF-α), interleukin 6 (IL-6), miRNA-34a, and miRNA-223, as well as a significant decrease in liver catalase (CAT) activity and in the levels of liver nuclear factor 1α (HNF-1α), sirtuin-1, and C/ebpα. Oral pretreatment with MESC (200 mg/kg) showed a significant decrease in caspase 2, CK18, TNF-α, IL-6 and a significant increase in liver CAT activity. MESC decreased the levels of liver miRNA-34a and miRNA-223 and induced HNF-1α, sirtuin-1, and C/ebpα gene expression. The histological examination showed a significant normalization in rats pretreated with MESC. Our findings showed that *Saussurea costus* may exert a potent hepatoprotective activity through the modulation of the expression of cellular cytokines, miRNA-34a, and miRNA-223.

## 1. Introduction

The use of medicinal plants for treating various ailments has been continuously explored and developed as an adjuvant to synthetic medicine; these plants also offer a broad range of bioactive secondary metabolites. *Saussurea costus* (syn. *Saussurea lappa,* Asteraceae) is considered one of the most important traditional Chinese medicinal plants. It is a rich source of various bioactive phytoconstituents, and its genus comprises about 300 species [1]. *S. costus* is generally known as *costus* root and Kust or Qist Hindi (Arabia). *S. costus* is traditionally used for treating asthma, gastric ulcer, inflammation, and liver diseases [2]. Many authors reported that its bitter roots have anxiolytic [3], anti-inflammatory, and antirheumatic activities [4]. *S. costus* was proved to be a potent inducer of apoptosis because of the presence of costunolide [5]. An aqueous extract of *S. costus* was found to be hypolipidemic. In Indian folklore, it was used as antidiabetic plant; recently, great efforts have been devoted to assessing its potential against diabetes [6]. Scientific evidence suggests its use as an antimicrobial and antiparasitic [5]. A methanol extract of the plant showed cardiotonic effects [7], while a petroleum ether extract showed potent anticonvulsant activity against picrotoxin-induced convulsions in mice [3]. *Saussurea* species contain sesquiterpene lactones, triterpenes, steroids, lignans, flavonoids, and some of these compounds possess interesting activities [8]. Sesquiterpenoids are the major constituents of the essential oil of *S. costus* roots, representing about 79.80% of its content [9]. The oil of *S. costus* is very expensive and has a very strong aroma; therefore, it used in high-grade perfumes and in preparations of hair oils, insecticides, and insect repellents [10]. Costunolide, a well-known sesquiterpene lactone isolated from *S. costus* roots, has antioxidant, anti-inflammatory, neuroprotective, and antidiabetic properties [11]. Acetylated flavone glycosides [12], palmitic, linoleic [5], and chlorogenic acids [13] were isolated from the *S. costus* roots. Because of the aforementioned diverse biological activities of *S. costus* and since it is an endangered species, India has prohibited its exportation either in crude form or in processed products [14].

According to WHO statistics (2018), Egypt ranks first in the world in liver disease deaths (68,866, representing 12.40% of total deaths) [15]; hence, our study focused on the hepatoprotective activity of *S. costus* roots. Paracetamol has long been established to cause liver toxicity, and its intentional abuse is used for suicide attempts [16]. Paracetamol is directly conjugated to glucuronic acid or sulfate and excreted into the bile. The remaining unconjugated paracetamol is then metabolized by P450 enzymes [17]. In humans and experimental animals, an acute or cumulative overdose of paracetamol can cause severe liver damage [18]. 

LC–MS/MS is a comprehensive analytical technology for the identification of plant metabolites [19]. In the literature, we found extensive studies performed on *S. costus* essential oil, while the chemical composition of the lipoidal matter or the plant phytoconstituents has not been studied in depth. Our current study attempted to explore *S. costus* various metabolites to which its medicinal activities may be attributed. UPLC/T-TOF–MS/MS was performed to analyze MESC, besides an analysis of lipoidal matter, as lipids are significant components of *S. costus*. Furthermore, MESC hepatoprotective activity was investigated by measuring the expression of miRNA-34A, miRNA-23, catalase, HNF-1α, sirtuin-1, C/ebpα in the liver as well as the plasma level of caspase 2, CK18, TNF-α, and IL-6.

## 2. Results and Discussion

### 2.1. GC/MS of MESC Lipoidal Matter

*S. costus* roots’ lipoidal matter contains a higher percentage of unsaturated fatty acids rather than that of saturated chain (84.48% and 15.52%, respectively). The major identified compounds were linoleic acid methyl ester, followed by oleic acid methyl ester (55.54% and 28.56%, respectively) (Table 1). 

### 2.2. UPLC/T-TOF–MS/MS

To obtain full coverage of the metabolomics of MESC, ESI–MS/MS was performed in negative and positive modes. The total ion chromatograms (TIC) of MESC are presented in Figure 1. The flavonoid aglycones *O*- and *C*-glycosides were identified; flavones and flavonol and their glycosides or glucuronic derivatives were the major components, reflecting the plant diverse biological activities [20]. Flavanones, isoflavones, and anthocyanidin glycosides or their derivatives, in addition to organic and phenolic acids and coumarins derivatives, were detected (Table 2, Figure 2 and Figure 3).

#### 2.2.1. Flavones

The glycoside diosmin (24) exhibited a protonated molecule [M+H]^+^ at *m/z* 609.1828; we also observed a subsequent loss of 146 Daltons for deoxyhexose at *m/z* 463.1245 and a loss of 308 amu for rutinoside at *m/z* 301.0719 [M+H-rutinose]^+^. Luteolin-*O*-hexoside (21) exhibited a deprotonated molecule [M-H]^−^ at *m/z* 447.0966; we then observed a subsequent loss of 162 amu for hexose at *m/z* 285.0520 [M-H-hexose]^−^, while luteolin-*C*-hexoside (31) showed a protonated molecule [M+H]^+^ at *m/z* 449.1549. Luteolin-di-*O*-hexoside (13) exhibited a deprotonated molecule [M-H]^−^ at *m/z* 609.1462; moreover, luteolin aglycone (40) was detected with a deprotonated molecule [M-H]^−^ at *m/z* 285.039, and characteristic peaks at *m/z* 269.1614 [M-H-OH]^−^ and 151.0980 indicated a dihydroxy-substituted A-ring [21].

Rhoifolin glycoside (Apigenin neohesperidoside) (20) showed a deprotonated molecule [M-H] at *m/z* 577.1506 and a fragment at *m/z* 269.0386 [M-H-neohesperidose]^−^. Apigenin-*C*-hexoside (22) showed a protonated molecule [M+H]^+^ at *m/z* 433.1232; however, apigenin-*O*-hexoside (32) showed a protonated molecule [M+H]^+^ at *m/z* 433.1851 and a fragment ion peak at *m/z* 271.1296 [M+H-hexose]^+^. Moreover, apigenin aglycone (34) was detected at *m/z* 269.0434 [M-H]^−^ [22]. Acacetin-*O*-rutinoside (10) showed a protonated molecule [M+H]^+^ at *m/z* 593.1631; on the other hand, acacetin aglycone (39) exhibited a deprotonated molecule [M-H]^−^ at *m/z* 283.1895 and a production at *m/z* 253.1469 corresponding to [M-H-OCH_3_]^−^. Baicalein-*O*-glycuronide (14) showed a protonated molecule [M+H]^+^ at *m/z* 477.0679.

#### 2.2.2. Flavonol

Myricetin (43) showed a protonated molecule [M+H]^+^ at *m/z* 319.1651 [23]. Gossypetin hexoside (30) was detected at *m/z* 481.1810 [M+H]^+^ and also showed a fragment at *m/z* 319.0663 [M+H-hexose]^+^. Quercetin hexoside (19) showed a protonated molecule [M+H]^+^ at *m/z* 465.1058; The neutral loss of 162 amu of the hexose moiety was observed in MS2 fragmentations at *m/z* 303.0471 [M+H-hexose]^+^. Quercetin glucuronide (11) was detected at *m/z* 477.0659 [M-H]^−^, and the neutral loss of 176 Daltons of the glycuronic moiety was confirmed by an ion at *m/z* 301.0332 [M-H-glycuronic acid]^−^. Quercetin pentoside (15) showed a deprotonated molecule [M-H]^−^ at *m/z* 433.1094; however, quercetin aglycone (28) was confirmed by an ion peak at *m/z* 303.0966 [M+H]^+^ [22]. Kaempferol glycuronide (23) showed a protonated molecule [M+H]^+^ at *m/z* 463.1023, and a neutral loss of 176 amu of the glycuronic moiety was detected at *m/z* 287.0589. The deprotonated molecule [M-H]^−^ of kaempferol neohesperidoside (16) was detected at *m/z* 593.1527. Isorhamnetin-*O*-hexoside (25) was detected by the presence of a protonated molecule [M+H]^+^ at *m/z* 479.1228, a production at *m/z* 317.0612 corresponding to [M+H-hexose]^+^, and a characteristic fragment at *m/z* 285.0323 [24]. The deprotonated molecule [M-H]^−^ of isorhamnetin-*O*-rutinoside (17) was detected at *m/z* 623.1595, the and a neutral loss of the rutinose moiety (308 amu) was detected at *m/z* 315.0418. 

#### 2.2.3. Flavanones and Isoflavones

Naringenin aglycone (36) showed a protonated molecule [M+H]^+^ at *m/z* 273.0884 and a characteristic fragment at *m/z* 147.0214 [M+H-C_6_H_6_O_3_]^+^ [25]. Hesperetin aglycone (41) showed a deprotonated molecule [M-H]^−^ at *m/z* 301.2041. Formononetin-*O*-hexoside (29) showed a protonated molecule [M+H]^+^ at *m/z* 431.1681 and a fragment at *m/z* 269.1243 [M+H-hexose]^+^. The deprotonated molecule [M-H]^−^ of formononetin aglycone (37) was detected at *m/z* 267.0712 and yielded a fragment due to the methyl group loss at *m/z* 252.0420 [26]. Daidzein-*C*-hexoside (18) showed a deprotonated molecule [M-H]^−^ at *m/z* 415.1944, while daidzein aglycone (27) was detected at *m/z* 255.0551.

#### 2.2.4. Flavanol and Anthocyanidin

Procyanidin B2 (12) showed a protonated molecule [M+H]^+^ at *m/z* 579.1859 and a fragment ion at *m/z* 561.1873 [M+H-H_2_O]^+^. Petunidin-*O*-hexoside (26) showed a protonated molecule [M+H]^+^ at *m/z* 479.0961 and a fragment ion at *m/z* 317.0662 [M+H-hexose]^+^. Malvidin-*O*-hexoside (33) was detected at *m/z* 491.1155 [M-H]^−^ and confirmed by a fragment at *m/z* 329.0696 for [M-H-hexose]^−^.

#### 2.2.5. Coumarins and Miscellaneous Metabolites

Tentatively identified coumarins were scopoletin (44), that was detected in both modes at *m/z* 191.0349 [M-H]^−^, and daphnetin (46), which was confirmed by a peak at *m/z* 179.0861 [M+H]^+^ with a fragment at *m/z* 163.0503. Aldehydes were tentatively identified as hexenal (47), showing a protonated molecule [M+H]^+^ at *m/z* 99.0437 and a fragment ion at *m/z* 83.0493, while cinnamaldehyde (48) showed a protonated molecule [M+H]^+^ at *m/z* 133.1001 and a fragment ion at *m/z* 117.0684 [27]. The chemotaxonomy marker of the genus *Costus*, diosgenin (saponin aglycone) (49), was also identified with a positive precursor [M+H]^+^ at *m/z* 415.3205, in accordance with published data [28,29].

#### 2.2.6. Amino, Organic, and Hydroxybenzoic Acid Derivatives

The common neutral loss of 44 Daltons was observed in MS2 fragmentations of phenolic and organic acids due to the loss of CO_2_ as in succinic and malic acids (at *m/z* 73.0277 and 89.0259, respectively) [30]. The identified amino acids were oxoproline, dimethylglycine, and 2-aminoadipic acid.

**Table 2 molecules-27-02802-t002:** Tentatively identified metabolites via UPLC/T-TOF–MS/MS in the methanol extract of *Saussurea costus* roots using negative and positive ionization modes.

No.	R_t_	Proposed Compounds	Formula	[M-H] m/z	[M+H] m/z	Diff. (ppm)	Ms^2^ (Characteristic Fragments)	%	Ref.
Amino acid derivatives									
1	1.22	Oxoproline	C_5_H_7_NO_3_	128.0348	130.0464	−0.7	84.0456	2.61	
2	2.51	Dimethylglycine	C_4_H_9_NO_2_		104.1061	2.4	59.0735, 58.0641	31.59	
3	9.39	2-Aminoadipic acid	C_6_H_11_NO_4_		162.0919	−2	161.0868	0.16	
4	12.97	N-alpha-acetyl-L-ornithine	C_7_H_14_N_2_O_3_		175.0754	−0.2	160.0846	0.12	
Organic and Hydroxybenzoic acid derivatives									
5	1.18	Succinic acid	C_4_H_6_O_4_	117.0188		−0.1	99.0094, 73.0277 [M-H-CO_2_^−^]^−^	0.78	[31]
6	1.23	Malic acid	C_4_H_6_O_5_	133.0141		−0.4	115.0052, 89.0259 [M-H-CO_2_^−^]^−^, 71.0145	2.32	
7	4.75	*p*-hydroxybenzoic acid	C_7_H_6_O_3_	137.0233		4.2	136.0153, 108.0218	1.49	
Hydroxycinnamic acid derivatives									
8	5.63	Chlorogenic acid	C_16_H_18_O_9_	353.0912	355.0752	4.2	191.0559 [M-H-caffeic acid]^−^, 161.0264 [M-H-quinic acid]^−^	2.66	[13]
9	13.59	3,4-dimethoxycinnamic acid	C_11_H_12_O_4_		209.0785	3.9	191.0686 [M+H-H_2_O]^+^	0.33	
Flavonoid derivatives									
10	5.17	Acacetin-*O*-rutinoside	C_28_H_32_O_14_		593.1631	−0.5	576.3024 [M+H-OH]^+^	1.08	
11	5.78	Quercetin glycuronide	C_21_H_18_O_13_	477.0659		3.1	301.0332 [M-H-glycuronic acid]^−^	0.13	
12	5.44	Procyanidin B2	C_30_H_26_O_12_		579.1859	−3	561.1873 [M+H-H_2_O]^+^	0.85	
13	6.52	Luteolin-di-*O*-hexoside	C_27_H_30_O_16_	609.1462		−3.9	563.2827	0.06	
14	6.83	Baicalein-*O*-glycuronide	C_21_H_18_O_11_		447.0679	0.6	412.2124, 268.1566	1.51	
15	6.88	Quercetin pentoside	C_20_H_18_O_11_	433.1094		4.6	326.9419	0.07	
16	7.11	Kaempferol neohesperidoside	C_27_H_30_O_15_	593.1527		−4.8	112.9892	0.09	
17	7.23	Isorhamnetin-*O*-rutinoside	C_28_H_32_O_16_	623.1595		1	315.0418 [M-H-rutinose]^−^, 299.0167	0.40	
18	7.25	Daidzein-*C*-hexoside	C_21_H_20_O_9_	415.1944	417.1285	2	355.0871, 238.0804	0.97	
19	7.33	Quercetin hexoside	C_21_H_20_O_12_		465.1058	−4	303.0471 [M+H-hexose]^+^	0.15	
20	7.43	Rhoifolin (Apigenin neohesperidoside)	C_27_H_30_O_14_	577.1506		2.9	531.2191, 269.0386 [M-H-neohesperidose]^−^	0.03	
21	7.44	Luteolin-*O*-hexoside	C_21_H_20_O_11_	447.0966		2.9	285.0520 [M-H-hexose]^−^	0.30	[32]
22	7.81	Apigenin-*C*-hexoside	C_21_H_20_O_10_		433.1232	0.7	403.1114, 151.0633	0.08	[33]
23	8.00	Kaempferol glycuronide	C_21_H_18_O_12_	461.0608	463.1023	2.7	287.0589 [M+H- glycuronic acid]^+^	1.28	
24	8.01	Diosmin (diosmetin-*O*-rutinoside)	C_28_H_32_O_15_		609.1828	−3.2	463.1245 [M+H-deoxyhexose]^+^, 301.0719 [M+H-rutinose]^+^	0.17	
25	8.04	Isorhamnetin-*O*-hexoside	C_22_H_22_O_12_	477.1412	479.1228	−4.1	317.0612 [M+H-hexose]^+^, 285.0323	0.27	
26	8.08	Petunidin-*O*-hexoside	C_22_H_23_O_12_		479.0961	−0.7	317.0662 [M+H-hexose]^+^	0.08	
27	8.14	Daidzein	C_15_H_10_O_4_		255.0551	−4.3	167.1071	0.31	
28	8.86	Quercetin	C_15_H_10_O_7_		303.0966	1.8	167.0781	1.99	[34]
29	9.09	Formononetin-*O*-hexoside	C_22_H_22_O_9_		431.1681	1.7	269.1243 [M+H-hexose]^+^	0.16	
30	9.15	Gossypetin hexoside	C_21_H_20_O_13_		481.1810	0.5	319.0663 [M+H-hexose]^+^	0.06	
31	9.43	Luteolin-*C*-hexoside	C_21_H_20_O_11_		449.1549	2.9	---	3.51	
32	9.46	Apigenin-*O*-hexoside	C_21_H_20_O_10_		433.1851	−2.6	271.1296 [M+H-hexose]^+^	0.49	[33]
33	10.41	Malvidin-*O*-hexoside	C_23_H_25_O_12_	491.1155	493.1319	0.8	329.0696 [M-H-hexose]^−^	0.98	
34	10.44	Apigenin	C_15_H_10_O_5_	269.0434		4	164.9932	0.16	[32]
35	10.80	3,5,7-trihydroxy-4’-methoxyflavone	C_16_H_12_O_6_	299.0541	301.0812	4.4	153.1010, 126.0668	0.13	
36	12.41	Naringenin	C_15_H_12_O_5_	271.1324	273.0884	0.6	257.0491, 147.0214 [M+H-C_6_H_6_O_3_]^+^	1.54	
37	12.71	Formononetin	C_16_H_12_O_4_	267.0712	269.0403	−1.9	252.0420 [M-H-CH_3_]^−^	0.27	
38	13.10	3’-methoxy-4’,5,7-trihydroxy flavonol	C_16_H_12_O_7_		317.1198	−4.8	302.0595, 300.1432, 275.0970	0.66	
39	13.63	Acacetin (5,7-Dihydroxy-4’-methoxyflavone)	C_16_H_12_O_5_	283.1895	285.0867	5	253.1469 [M-H-OCH_3_]^−^	11.43	
40	13.71	Luteolin	C_15_H_10_O_6_	285.039	287.0956	−0.3	269.1614 [M-H-OH]^−^, 151.0980	1.23	[35]
41	14.20	Hesperetin	C_16_H_14_O_6_	301.2041	303.1557	1	240.0945, 141.0669	0.09	
42	15.64	4’,5-dihydroxy-7-methoxyflavone	C_16_H_14_O_5_		287.1053	−1.7	213.1243	1.72	
43	17.63	Myricetin	C_15_H_10_O_8_		319.1651	2.8	----	0.24	[23]
Coumarins									
44	9.08	Scopoletin	C_10_H_8_O_4_	191.0349	193.1578	0.6	176.0147, 148.0164	11.82	[32]
45	16.74	Esculetin hexoside	C_15_H_16_O_9_	339.1987		1.9	---	0.03	
46	17.74	Daphnetin	C_9_H_6_O_4_		179.0861	0	163.0503	0.19	
Miscellaneous metabolites									
47	3.21	Hexenal	C_6_H_10_O		99.0437	1	83.0493	0.36	
48	17.27	Cinnamaldehyde	C_9_H_8_O		133.1001	0.7	131.0851, 117.0684	0.58	
49	20.98	Diosgenin	C_27_H_42_O_3_		415.3205	1.89	283.2412	0.08	[28,29]

### 2.3. In Vivo Hepatoprotective Activity

In the present study, MESC was evaluated for its hepatoprotective activity in paracetamol-induced liver toxicity. Plasma caspase 2 and cytokeratin 18 (CK18) were elevated in the group Gp II in comparison with the control group. Hofer et al. reported that caspase-cleaved cytokeratin 18 (CK18-Asp396) values were increased significantly in Gp II [36]. Caspase-2 is an activator caspase involved in a number of apoptotic pathways, remarkably, in response to intracellular stress factors (e.g., DNA damage, ER stress). Caspase-2 is found in injured hepatocytes, and its activity has indeed been found to be strongly raised in non-alcoholic steatohepatitis patients and mouse models [37]. CK18 is the most widely available intermediate filament protein in the liver and helps caspase substrates during hepatic cells apoptosis. High CK18 levels have been found in hepatocellular carcinoma, viral liver infections, alcoholic liver disease, NAFLD, and cholestatic liver disease [38].

Toxicity by paracetamol (1 g/kg) led to a significant elevation (*p* < 0.01) in plasma caspase 2 and CK18 levels, corresponding to 346.39% and 139.31% increases, respectively, as compared to the levels in Gp I, considering the values in the control group as 100% (Table 3). Pretreatment with silymarin significantly decreased plasma caspase 2 and CK18 levels by 42.36% and 29.38%, respectively compared to the levels in Gp II. Pretreatment with MESC significantly decreased the level of plasma caspase 2 and CK18 by 54.93% and 23.17%, respectively compared to those in Gp II. MESC and silymarin pretreatment reduced the levels of caspase 2 and CK18 compared with those in Gp II. The depletion of caspase 2 and CK18 was demonstrated to enhance apoptosis and decrease cell viability by affecting caspase-2 activity in hepatocellular carcinoma cells [39].

Toxicity by paracetamol led to a significant increase in liver TNF-α and IL-6 levels by 294.70% and 217.35%, respectively, as well as a significant decrease (*p* < 0.01) in liver CAT level by 50.37%, compared to the levels in Gp I (Table 3). Pretreatment with silymarin significantly decreased liver TNF-α and IL-6 levels by 51.69% and 54.12%, respectively, as well as induced a significant elevation in liver CAT level by 151.59%, compared to the levels in Gp II. Pretreatment with MESC significantly decreased the levels of liver TNF-α and IL-6 by 65.22% and 52.06%, respectively, as well as significantly increased liver CAT level by 164.73% compared to the levels in Gp II.

Paracetamol-induced liver toxicity and cell inflammation led to infiltration of inflammatory cells and to inflammatory cytokine upregulation (TNF-, IL-1, and IL-6), resulting in inflammation [40,41]. MESC and silymarin pretreatment suppressed the elevation of serum TNF-α and IL-6, reducing their inflammatory effect and inducing CAT activity in the liver of Gp II rats. Therefore, the hepatoprotective effect of MESC was related to its anti-inflammatory activity. 

A significant decrease of the expression of HNF-1α, sirtuin-1, and C/ebpα genes by 55.10%, 61.22%, and 77%, respectively, as well as a significant increase of the expression of liver miRNA-34a and miRNA-223 genes by 447.27% and 344.79%, respectively, in Gp II rats as compared with Gp I rats (*p* < 0.05) were observed (Figure 4 and Figure 5). Silymarin administration showed a significant increase (*p* < 0.05) in C/ebpα, HNF-1α, and sirtuin-1 gene expression and a significant decrease (*p* < 0.05) of liver miRNA-34a and miRNA-223 genes expression compared to the levels in Gp II. MESC administration showed a significant increase (*p* < 0.05) in the expression of liver C/ebpα, HNF-1α, and sirtuin-1 genes by 404.35%, 179.55%, and 223.64%, respectively, as well as a significant decrease (*p* < 0.05) of liver miRNA-34a and miRNA-223 genes expression by 46.27% and 40.49%, respectively, compared with the levels in Gp II. Paracetamol overdose could suppress C/ebpα gene expression in hepatocytes in vivo, which was accompanied by a significant accumulation of cytokines. MESC and silymarin protected rats against paracetamol-induced hepatotoxicity. Oxidative stress due to C/ebpα gene expression depletion and uncontrolled ROS is known to be the primary pathogenic mechanism of paracetamol-induced liver toxicity, as well as the main inhibitor of hepatocytes protective factors. Our metabolomics results revealed that MESC is rich in flavonoid compounds that exerted pronounced antioxidative effects against paracetamol-induced depletion of C/ebpα gene expression in the hepatocytes of Gp II rats [42]. 

HNFs were discovered to be liver-enriched transcription factors with various functions in the transcription of liver-specific genes [43]. Although HNF-1 affects many organs, hepatic responses are the most noticeable due to its high liver concentration and local production. [44]. Inactivation of HNF-1 has been linked to a variety of tumor-promoting pathways [44]. In our study, HNF-1α gene expression was significantly decreased compared to the control levels after paracetamol administration. No data are available about the direct effect of paracetamol administration on the fold changes in HNF-1α gene expression. Our study suggests that the inactivation of HNF-1α gene expression may be due to the elevation of TNF-α in paracetamol-treated rats. A study by Bao et al. implicated TNF-α-induced inhibition of HNF-1α in the promotion of HCC disease progress [45]. MESC and silymarin pretreatment suppressed TNF-α production, which might lead to the induction of HNF-1α expression. Flavonoids have the potential of exerting positive health benefits and increase the expression of HNF-1α [46,47].

Sirtuin-1 has a role in many biological processes such as energy metabolism, apoptosis, and inflammation. Our results showed a significant decrease in liver sirtuin-1 gene expression in paracetamol-treated rats, in agreement with Wojnarová et al. [48]. MESC and silymarin pretreatment activated the expression of liver sirtuin-1 in Gp II rats. Many studies reported the activation of liver sirtuin-1 gene expression after treatment [48,49]. 

Our data showed an increase in hepatic miRNA-34a and miRNA-223 gene expression in Gp II rats. miRNA-34a was shown to be increased in hepatic fibrosis, HCV infection, alcoholic liver disease, and in the presence of cardiotoxicity [50,51]. Inhibiting miRNA-34a suppressed lipid accumulation and improved the degree of steatosis. Additionally, silencing miRNA-34a led to an increase in the expression of sirtuin-1, regulating the activity of AMP kinase [52]. miRNA-223 is elevated in the serum of patients with hepatitis B and HCC, implying that miRNA-223 could be used as a new biomarker for liver injury. [53]. MESC pretreatment induced the downregulation of hepatic miRNA-34a and miRNA-223 gene expression in Gp II rats. The above results indicated that miRNA-34a and miRNA-223 are components of very important mechanism underlying the treatment of liver toxicity by MESC. 

Liver sections of the control group (Gp I) demonstrated normal hepatocytes with no fibrosis or inflammation (Figure 6a), while liver tissue of Gp II showed degenerated hepatocytes (hydropic degeneration, yellow arrows in Figure 6b) with fibrosis (Figure 6b). The hepatocytes of Gp III rats were normal, without fibrosis or inflammation after silymarin pretreatment (Figure 6c). Additionally, the hepatocytes showed recovery without hydropic degeneration in MESC-pretreated rats (Gp IV) as compared with paracetamol-treated rats (Figure 6d). According to these histological studies, MESC and silymarin have a hepatoprotective effect against paracetamol-induced liver toxicity.

Our LC–MS/MS results showed that flavonoids were the major components of *S. costus*; which supports the hepatoprotective activity of this plant [54]. Coumarins as well possess potent antioxidant activity and are therefore regarded as potent hepatoprotective compounds [55]. Tejaswi et al. proved the anti-inflammatory activity of *S. lappa* root extract in rats [54]; moreover, ALT, AST, total protein and albumin levels showed an improvement after the extract administration in rabbits [56,57]. This study showed for the first time an improvement in all measured genetic, apoptotic markers and inflammatory cytokines after *S. costus* root extract pretreatment.

## 3. Materials and Methods

### 3.1. Chemicals

Methanol (HPLC grade), paracetamol (99%), silymarin (98.5%), and Tween 80 were purchased from Merk (Zug, Switzerland). Other reagents were of high analytical grade.

### 3.2. Plant Materials and Extraction Process

*Saussurea costus* (Falc.) Lipsch. roots were purchased from a local herbalist (Giza, Egypt, 2020). Identification of the plant was performed by the Agriculture Research Center, Cairo, Egypt. In Soxhlet, powdered roots (1 kg) were extracted with methanol (4 × 500 mL, 1 h). The extract was left to cool, filtered, and then evaporated (Rotavapor^®^, BÜCHI, Flawil, Switzerland) [58]. The obtained dried extract was used for biological and chemical investigations.

### 3.3. Gas Chromatography–Mass Spectrometry Analysis (GC–MS) of the Lipoidal Matter

Powdered roots (100 g) were macerated in *n*-hexane (2 × 100 mL), filtered, and evaporated. The crude oil was investigated for its composition in a GC–MS system (Agilent Technologies, Santa Clara, CA, USA) equipped with a mass spectrometer detector (5977A) at the National Research Centre, Cairo, Egypt. The GC was equipped with an HP-5MS column. 

### 3.4. UPLC/T-TOF–MS/MS Analysis

The MESC was analyzed at the Proteomics and Metabolomics unit of the Children’s Cancer Hospital Egypt 57357, Cairo, Egypt. The analysis was carried out using an Exion LC Triple TOF 5600+ system (SCIEX, Framingham, MA, USA) operated at 40 °C and equipped with an X select HSS T3 C-18 column (Waters Corporation, Milford, CT, USA, 2.5 μm, 2.1 × 150 mm) and a precolumn (Phenomenex, In-Line filter disks, 0.5 μm × 3.0 mm). MESC (50 mg) was dissolved in solvent working solution (MilliQ water: methanol: acetonitrile–50:25:25), sonicated (10 min), then centrifugated (10,000 rpm, 10 min). The stock sample (50 µL) was diluted with the working solvent (1000 µL). The phytoconstituents of MESC were analyzed using UPLC/T-TOF–MS/MS in both negative and positive modes [59]. Samples (1µg/µL, 10 µL) were injected using the following mobile phases: solvent A was ammonium formate buffer (5 mM, pH 8 using NaOH) containing methanol (1%), for the negative mode, while in the positive mode, solvent A was ammonium formate buffer (5 mM, pH 3 using formic acid) containing methanol (1%), and in both modes, solvent B was acetonitrile (100%). The gradient elution was performed as follows: isocratic 90%: 10% (0–1 min), linear from 90%: 10%, to 10% and 90% (1.1–20.9 min), isocratic 10%: 90% (21–25 min), and finally isocratic 90% and 10% (25.1–28 min) of solvent A and B, respectively. The flow rate was 0.3 mL/min. The working solvent (10 µL) was injected as a blank sample. The detected phytoconstituents were recorded by Analyst TF 1.7.1 software, Peak view 2.2 software (SCIEX, Framingham, MA, USA) and MS-DIAL 3.70 software for data processing [60]. Mass spectrometry (MS) was done on a Triple TOF 5600+ system equipped with a Duo-Spray source operating in the ESI mode (AB SCIEX, Framingham, MA, USA) ranged from 50 to 1100 *m/z*. The characterization of compounds was performed by the generation of the candidate formula with a mass accuracy limit of 10 ppm and also considering Rt, MS2 data, databases, and reference literature [61]. 

### 3.5. In Vivo Hepatoprotective Activity

#### 3.5.1. Experimental Design

Adult male albino rats (150 ± 10 g, 40) were purchased from the National Cancer Institute, Cairo University, Giza, Egypt. Rats were provided water and standard diet ad libitum, observed daily, and kept in polypropylene cages under normal environmental conditions (22 ± 2 °C). The prophylactic potential of MESC against paracetamol hepatotoxicity was evaluated. Ethical approval was obtained by the ethic committee of the faculty of applied medical science (20210802, 2 August 2021). Animals and the treatment schedule (4 weeks) were as follows: Gp I (normal control) and Gp II (positive control); rats were orally given Tween 80 (3 mL, 1%) in saline daily; Gp III rats were orally treated with silymarin (50 mg/kg, in 1% tween 80), daily [62]. Gp IV rats were orally treated with MESC (200 mg/kg, in 1% tween 80) daily [63]. On day 27, the animals of all groups fasted for 18 h. On day 28, all animals in Gp II, III, and IV received paracetamol (1 g/kg) [64]. On day 29, blood samples from a retro-orbital vein were collected in heparin-containing tubes and centrifuged (1000 rpm, 20 min). Plasma was collected, and caspase 2 and CK18 were evaluated using the ELISA technique (BMG Labtech Spectrostar Nano, Ortenberg, Germany). The Caspase-2 ELISA kit is based on the competitive enzyme immunoassay technique, which employs a monoclonal anti-Caspase-2 antibody and a Caspase-2-HRP conjugate. In a microplate reader, the intensity of color is spectrophotometrically measured at 450 nm. The CK18 kit employs the Double Antibody Sandwich Technique, which relies on the antigen characteristics with more than two valances and can recognize the coated antibody and the detection antibody at the same time. The liver tissue was homogenized and centrifuged in phosphate buffer (3 mL, pH 7.2; 3000× *g*, 10 min). The clear supernatant was tested for CAT using a Cayman Chemical Company kit (An Arbor, MI, USA). The UV spectrophotometric method for measuring catalase activity is centered on monitoring the change in 240 nm absorbance at high concentrations of hydrogen peroxide (30 mM). TNF and IL-6 levels in the liver were measured using a microplate reader at 450 nm (Thermo Electric Corp., Shanghai, China). An anti-IL-6 or an anti-TNF-α monoclonal antibody and a biotin-conjugated monoclonal anti-IL-6 or anti-TNF-α antibody that binds to IL-6 or TNF-α captured by the first antibody were used in an indirect sandwich enzyme-linked immunosorbent assay to evaluate IL-6 and TNF-α. Streptavidin–HRP was added to the wells, followed by a substrate solution reacting with HRP. After stopping the reaction with acid, the absorbance was measured.

#### 3.5.2. Real-Time PCR

The TRIzol method was used to isolate total RNA from the rats’ liver (Life Technologies Corp., Grand Island, NY, USA). RNA (1 μg) in reaction buffer was mixed with dithiothreitol (10 nmol/L), oligo (dT) primer (25 pg), 0.5 mmol/L of each deoxyribonucleoside triphosphate (dNTP), and 200 units superscript II Rnase H-Reverse Transcriptase. The reactions were kept at 42 °C for 2 min, 42 °C for 50 min, 70 °C for 15 min, and then chilled to 4 °C. The PCR reaction mixture consisted of PCR buffer, MgCl_2_, (1.5 mM), 0.2 mM of each dNTP, 0.4 μM of specific primers (Table 4) for hepatocyte nuclear factor 1α (HNF-1α), sirtuin-1, CCAAT-enhancer-binding proteins (C/ebpα), miRNA-34a, and miRNA-223. The PCR reaction mixtures were incubated at 94 °C for 3 min, then continued for each molecule to be analyzed with the respective number of cycles at 94 °C for 45 s and at their corresponding annealing temperatures for 30 s, followed by 1 min 30 s at 72 °C. Following that, a 10 min extension step at 72 °C was performed. β-Actin mRNA was used as a housekeeping gene to normalize the CYP mRNA content [65]. 

#### 3.5.3. Histological Assessment

The liver pieces were fixed in formaldehyde solution (10%) and then were examined for histopathological changes.

### 3.6. Statistical Analysis

Statistical analyses were carried out using GraphPad Prism (version 5.01, San Diego, CA, USA). ANOVA with posttest Tukey’s multiple comparisons were performed. Values (n = 10) are presented as mean ± standard deviation for ELISA measurements and PCR analyses of gene expression (*p* < 0.05 was considered statistically significant). 

## 4. Conclusions

Herein for the first time, we were able to identify the chemical profile of *Saussurea* *costus* root through LC–MS analysis. The tentatively identified phytoconstituents (49) were characterized as belonging to different chemical classes, flavonoids being the major constituents besides organic acids, coumarins, and anthocyanidin. *S. costus* has a potent hepatoprotective activity through the modulation of cellular cytokines release and miRNA-34a and miRNA-223 expression. *S. costus* reduced the levels of caspase 2 and CK18, TNF-α, and IL-6 and increased catalase activity. Furthermore, it increased HNF-1α, sirtuin-1, and C/ebpα expression and decreased miRNA-34a and miRNA-223 gene expression. *S. costus* minimized some negative symptoms and pathological changes. We hope that this study will attract attention towards this plant myriad chemical constituents and its great potential in health care.

## Figures and Tables

**Figure 1 molecules-27-02802-f001:**
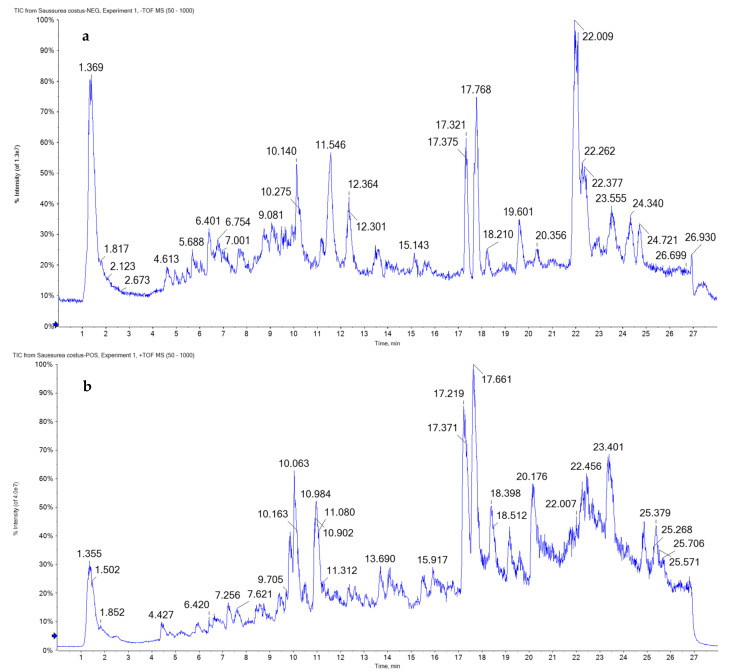
UPLC/T-TOF–MS/MS chromatograms of methanol extract of *Saussurea costus* roots in negative (**a**) and positive (**b**) ionization modes.

**Figure 2 molecules-27-02802-f002:**
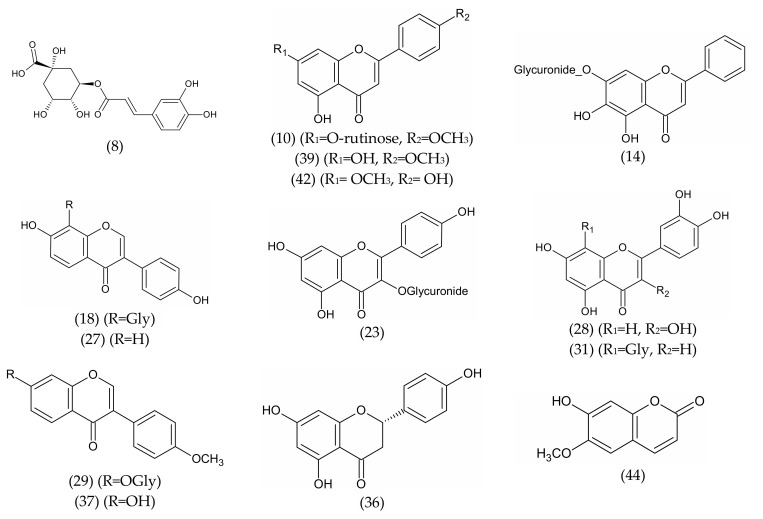
Representative structures of the major metabolites (flavonoids, phenolics, and coumarins) identified in the methanol extract of *Saussurea costus* roots.

**Figure 3 molecules-27-02802-f003:**
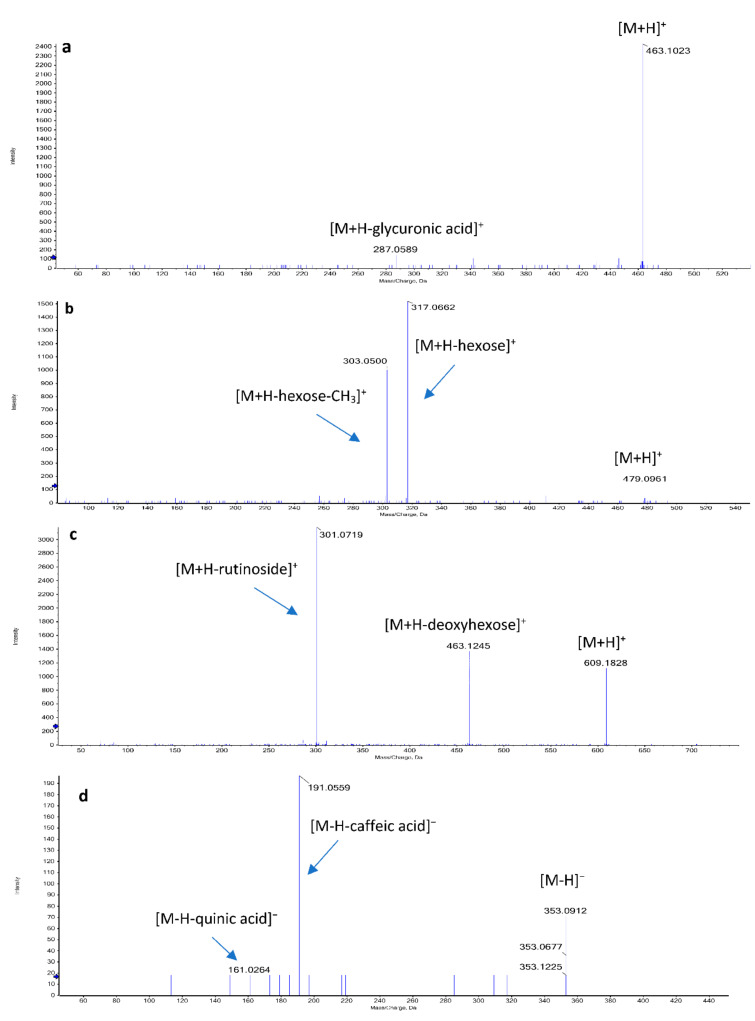
Mass fragments of some identified phytoconstituents in the methanol extract of *Saussurea costus* roots: (**a**) kaempferol glycuronide, (**b**) petunidin-*O*-hexoside, (**c**) diosmin, (**d**) and chlorogenic acid.

**Figure 4 molecules-27-02802-f004:**
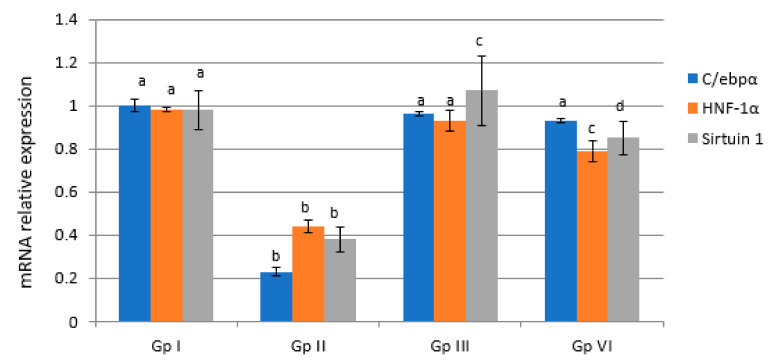
Effect of MESC and silymarin on the expression levels of liver C/ebpα, HNF-1α, and sirtuin-1 genes in paracetamol-treated rats. Data are presented as fold increases with respect to control values, considering the normal control values equal to 1. Data followed by the same letter within the same parameter are not significantly different at *p* ≤ 0.05.

**Figure 5 molecules-27-02802-f005:**
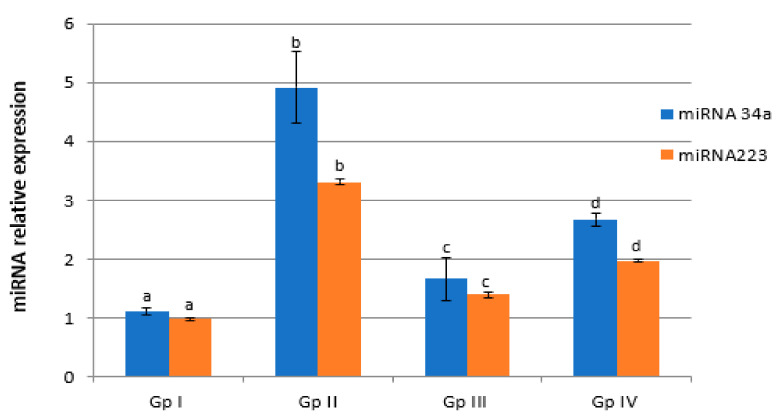
Effect of MESC and silymarin on the expression levels of liver miRNA-34a and miRNA-233 gene in paracetamol-treated rats. Data are presented as fold increases with respect to control values, considering the normal control values equals to a. Data followed by the same letter within the same parameter are not significantly different at *p* ≤ 0.05.

**Figure 6 molecules-27-02802-f006:**
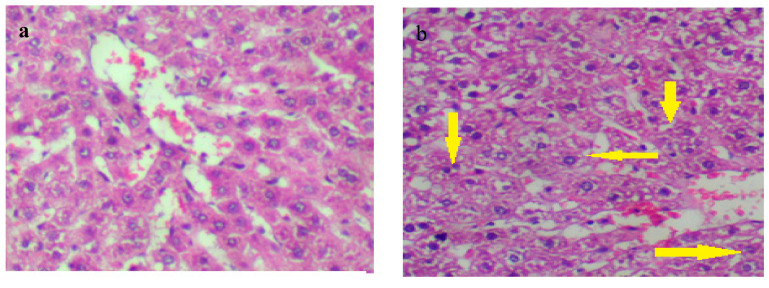
Tissue sections stained with hematoxylin and eosin (H&E; 400×) for the histological examination of the liver tissues of different groups in comparison to the control group (**a**) Gp I: normal control, (**b**) Gp II, (**c**) Gp III: (silymarin 50 mg/kg), (**d**) Gp IV: (MESC 200 mg/kg).

**Table 1 molecules-27-02802-t001:** Results of the GC/MS analysis of the lipoidal matter derived from *Saussurea costus* roots.

Peak	R_t_	Name	Formula	%
1	5.595	Heptyl hydroperoxide	C_7_H_16_O_2_	0.07
2	11.274	Dihydromyrcene (Citronellene)	C_10_H_18_	0.06
3	13.709	Palmitic acid methyl ester	C_17_H_34_O_2_	9.73
4	17.123	Linoleic acid methyl ester	C_19_H_34_O_2_	55.54
5	17.22	Oleic acid methyl ester	C_19_H_36_O_2_	28.56
6	17.485	Methyl stearate	C_19_H_38_O_2_	5.42
7	20.67	2-pentadecynyl alcohol	C_15_H_28_O	0.19
8	21.201	Cyclopentaneundecanoic acid, methyl ester	C_17_H_32_O_2_	0.30
9	26.863	*Cis*-3-Hexenylpyruvate	C_9_H_14_O_3_	0.13

**Table 3 molecules-27-02802-t003:** Effect of MESC and silymarin on plasma caspase 2, cytokeratin 18 (CK18), and liver TNF-α, interleukin 6 (IL-6), and catalase (CAT) in experimental rats.

Gps	Caspase 2 (pg/mL)	CK18 (U/L)	TNF-α (ng/g Tissue) ×1000	IL-6 (pg/mL)	CAT (cat/g Protein)
Gp I	72.92 ± 6.76 ^a^	646.23 ± 22.61 ^a^	3.21 ± 0.20 ^a^	41.97 ± 3.49 ^a^	68.14 ± 5.33 ^a^
Gp II	252.59 ± 12.53 ^b^	900.29 ± 19.79 ^b^	9.46 ± 1.16 ^b^	91.22 ± 4.50 ^b^	33.80 ± 3.21 ^b^
Gp III	145.59 ± 10.50 ^c^	635.76 ± 2.95 ^c^	4.57 ± 0.37 ^c^	41.85 ± 2.99 ^a^	51.24 ± 4.81 ^c^
Gp IV	113.84 ± 10.36 ^c^	691.70 ± 21.70 ^c^	3.29 ± 0.24 ^a^	43.73 ± 3.91 ^a^	55.68 ± 4.58 ^c^

Data shown are mean ± standard deviation (n = 10). Data followed by the same letter within the same parameter are not significantly different at *p* ≤ 0.05.

**Table 4 molecules-27-02802-t004:** Primers used in real-time PCR.

Gene	Primer Sequence
HNF-1α	F: 5′-GACCTGACCGAGTTGCCTAAT-3′ R: 5′-CCGGCTCTTTCAGAATGGGT-3′
Sirtuin-1	F: 5′- CCAGAACAGTTTCATAGAGCC-3′ R:5′-TCTTACTTTCAGAGAAGACCCAATA-3′
C/ebpα	F:5′- CAAGAACAGCAACGAGTACCG-3′ R: 5′- GTCACTGGTCAACTCCAGCAC-3′
miRNA-34a	F:5′--UGG CAG UGUCUU AGC UGG UUG UU-3′ R: 5′-CAA CCAGCU AAG ACA CUG CGA AA-3′.
miRNA-223	F: 5′-TGGATCCGTGTCACTCGGGCTTTACCTG-3′ R: 5′- CGAATTCGTAGACACAGCCCAGGGCTGT-3′.
β-Actin	F:5′-GGCTGTATTCCCCTCCATCG-3′ R: 5′- CCAGTTGGTAACAATGCCATGT-3′

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
