# Peer review of "Tentatively Identified (UPLC/T-TOF–MS/MS) Compounds in the Extract of Saussurea costus Roots Exhibit In Vivo Hepatoprotection via Modulation of HNF-1α, Sirtuin-1, C/ebpα, miRNA-34a and miRNA-223"

_molecules, 2022, doi:10.3390/molecules27092802_

Round 1
Reviewer 1 Report
Since about 2017 S. costus was tested in vivo on animal models of thyroid, hepatic, renal, spleen and pulmonary disorders and some therapeutic effects of its products were confirmed. The research fit well with this important trend of pre-clinical research of natural products to provide evidence for their usefulness. Thus, the idea of the manuscript is interesting from both, scientific and practical point of view. However, I have some comments and questions, which need to be addressed before the acceptance of the manuscript to be published in Molecules.
Major comments:
- Abstract: Could you explain why you wrote: “For the first time, we were able to identify the major metabolites…”, while the most of biologically active metabolites of costus are well known?
- Abstract: “The hepatoprotective activity of MESC was evaluated using ELISA and PCR techniques” – What about the methods used to evaluate CAT, TNF-α and IL6?
- Introduction: The statement: “Herbal medicine is regarded as safe to be used …” is controversial. Many of herbal products are known as both medicaments and poisons, depending, for example, on the concentration or application form. Consider and improve the statement.
- Introduction: “ costus is traditionally used in…” – the list of ailments is much wider (even in the reference you gave: cough, pulmonary disorders, chronic skin diseases, infections, rheumatism). Expand the list.
- Introduction: The sentence “Hepatotoxicity by paracetamol has been related to several cases of cirrhosis, hepatitis, and suicide attempts” is unclear. Does abuse of paracetamol cause cirrhosis and hepatitis, and how is it related to suicide attempts?
- Introduction: “additionally, hepatoprotective activity was also investigated” – Additionally? I expect it was one of the main aim of this study (“Egypt ranks the first in the world in liver disease deaths”).
- Some important information in Introduction are missing, e.g. the role of caspase 2 or cytokeratin 18 in hepatocyte apoptosis / necrosis during liver diseases, which would explain why did you test their concentration. Complete shortly and give more recent references except ref. no 36, 37.
- Explain the differences between Gp I rats (normal control) and Gp II rats (positive control) (Your only description to both groups: “rats were given tween 80 (3 mL, 1%) in saline orally daily”) – precise this description. Use the abbreviations Gp I / Gp II in the description of the results to provide a precise information to which group the results were compared.
- Results: If you use the percentages comparing the results (e.g. “significant increase (P< 0.01) in plasma caspase 2 and CK18 levels to 346.39 %...as compared to the control group”) you have to explain in description whether the concentrations in control were considered as 100%.
- Result: Based on your study the statement: “MESC could prevent liver toxicity via blocking of cytokines activity” is imprecise and it can be confusing (suggestion: MESC may bind to the cytokines blocking their activity). You did not check the cytokine activity, but their concentration. Improve.
- Figures 4 and 5: shift of the graph with the description of statistics results (possible, that it happened during submission); lack the description of Y axis
- Results, line 108: What does it mean “developmental pathways”?
- Results: “Our results come in accordance with a study done by Tejaswi et al., which proved the hepatoprotective activity of Saussurea lappa root extract [54] – see also http://dx.doi.org/10.22159/ajpcr.2019.v12i8.34218
- Methods: The lack of ELISA and other methods description. The sentences: “Plasma was collected and caspase 2 and CK18 were evaluated using ELISA techniques …assessment of CAT colorimetrically using a kit …liver TNF-α and IL6 were detected using a UV microplate reader” are far insufficient. Give precise information about these methods (e.g. kind of ELISA kits, principles of each method, marking of TNF-α and IL-6 for detection).
- Conclusion: “Our study on costus plant gives a new challenge to eradicate liver diseases” – The statement is ineligible. Using plant preparations you cannot eradicate liver disease (e.g. hepatitis caused by HBV/HCV or alcoholic cirrhosis), you can minimize some negative symptoms / pathological changes. What is “costus”?
Minor comments:
- Graphical Abstract: Because of in vivo experiments were done on white rats, replace the mice (figures/ photos) on appropriate animals.
- Whole text: unify the term UPLC/Triple-TOF-MS/MS instead of UPLC/T-TOF
- Whole text: the spelling rule for interleukin abbreviations with a hyphen: IL-6, IL-2. Improve.
- Introduction: “syn;” – probably dot should be instead of semicolon; “(Arab)” – Arabian / Arabic / Arab.
- Introduction: “Sesquiterpenoids were the major content … than monoterpenoids…” – If something is the major content, for sure its concentration is higher than other constituent. Improve stylistically.
- Introduction: “…hence our study targets the hepatoprotective activity of costus roots. Hepatotoxicity by paracetamol …” – Write otherwise this fragment not to mix hepatoprotective role with hepatotoxicity
- Results and whole text: add coma before “respectively”
- Table 3: improve units: pg/mL instead of Pg/mL, cat/g protein instead of katal/g, remove the double dot in the description
- Results, line 60-61: The sentence “MESC and silymarin pretreatment improve the levels of caspase 2 and CK18 compared with paracetamol-treated rats.” is completely unnecessary. The more that, in this form suggests elevated level of CAS2 and K18 after MESC and silymarin pretreatment.
- Results and whole text: unify the shortcut of rat groups – small / capital letters (Gp versus gp)
- Conclusion: “through modulating cellular cytokines” what? (production / release)
- Conclusion: “Saussurea costus improves the levels of caspase 2 and CK18” – Improve the sentence not to suggest the elevated level of CAS2 and K18
Reviewer 2 Report
The article entitled „Tentatively Identified (UPLC/T-TOF-MS/MS) Extract of Saussurea costus roots exhibits in vivo Hepatoprotection via Modulation of HNF-1α, Sirtuin-1, C/ebpα, miRNA-34a and miRNA-223” presents an interesting work on secondary metabolites responsible for hepatoprotective action of plant traditionally used in folk medicine. The authors were able to identify the major metabolites, such as flavonoids, phenolic acids, coumarins, and stilbenes. The hepatoprotective activity of plant extract was evaluated using ELISA and PCR techniques. Authors’ findings showed that Saussurea costus has a potent hepatoprotective activity through modulating cellular cytokines, miRNA-34a and miRNA-223 expression.
In my opinion article is worth publishing in Molecules, however small correction are needed.
- Nomenclature of some compounds, e.g. “3,4-dimethoxy cinnamic acid” should be written together, not separately “3,4-dimethoxycinnamic acid”, other names for this type of compound as well. The letter "C" for C-glycosides should be italicized like "O" for O-glycosides. “P” in the name of para isomer should be written with a lowercase letter (p-hydroxybenzoic acid) etc. Please, correct.
- Figures – please, increase font size on the Figure 3. Figures 4 and 5 - the letters a, b, and c are in the wrong places - please correct.
- Table 2. – Please, label correctly – negative mode with “-“ and positive mode with “+” in superscript outside the square brackets.
Reviewer 3 Report
The paper describes the chemical composition of the root extract from Saussurea costus as well as its hepatoprotective action. Some corrections listed below should be done in the paper. Additionally, the differences between this paper and reference 54 must be specified. The novelty of the paper must be clearly highlighted.
Lines 97 & 98: authors found two apigenin derivatives with almost similar [M+H]+= 433.1. However, these compounds were attributed to apigenin-C- & apigenin-O-hexoside. Please explain the presence of C-and O-derivatives.
Line 111: glucuronic moiety instead glycuronic moiety.
Item 2.2. described the same data shown in table 2 without any additional information. The text seems unnecessary.
The presence of stilbenes in Saussurea and Asteraceae is unlikely. The detection of this chemical class should be investigated deeply by the authors.
Figure 3 should be transferred to supplementary material. Additionally, the mass spectra should be shown for all identified compounds exhibited in Fig 3.
Figures 4 & 5 must be redone. The letters of statistical analysis are not localized.
Lines 166 & 167: The study of Tejaswi et al. (reference 54) proved the hepatoprotective activity of Saussurea lappa root extract, according to the text written by the authors. On the other hand, in lines 35 & 36 authors affirmed that S. costus and S. lappa are synonymous. Therefore, my question is: What are the new data and the novelty of this paper?
Round 2
Reviewer 1 Report
Almost all of my comments have been thoughtful, explained and included in revised text. I still have one comment, resulting from a misunderstanding of my previous main remark no.10. Once again: in original text you wrote that “MESC could prevent liver toxicity via blocking of cytokines activity”, so it was your default suggestion, that MESC may for instance bind to the cytokines blocking their activity. However, because you did not check the cytokine activity, but only their concentration, you cannot write such a conclusion. Think it over, improve the sentence or delete it. Thus, in my opinion the manuscript can be published in Molecules after minor revision.
Reviewer 3 Report
Authors carried out the corrections (at least one major part) suggested by this reviewer.
Author Response
there are no comments from reviewer
Authors carried out the corrections (at least one major part) suggested by this reviewer.
This manuscript is a resubmission of an earlier submission. The following is a list of the peer review reports and author responses from that submission.
Round 1
Reviewer 1 Report
The authors have substantially improved the quality of the manuscript, however, some proposed points need to be revised.
In line 140 the authors address the characterization of Miscellaneous metabolites, however they write: "showed a protonated molecule [M+H]+ at m/z 133.1001 and the loss of 16 amu due to hydrolysis of an oxygen atom from the aldehyde group".
It is necessary to clarify the proposed fragmentation mechanism. I consulted the cited reference [24] and saw nothing about the proposed fragmentation.
Reviewer 2 Report
This manuscript is very interesting and relevant because it shows the compositional data of Saussurea costus roots and their hepatoprotective effects. The work is original. Sophisticate equipment was used. My specific comments are:
-Line 12 might be rewritten. Perhaps “Our study accomplished the analysis of the UPLC/T-TOF-MS/MS analysis of…”
-The authors should search for other keywords. The keywords they are using are already in the title.
-The introduction section should explicitly include the justification of the work. The authors should explain with more details the compositional data already known for roots of this plant, indicating what is still missing or why they considerer that the composition of this plant is not completely known. The analysis of lipids in this plant should also be justified in the introduction section.
-Please, explain what “reagents unmarked” means (line 153). I was wondering if you might find another term for that.
-Please, give the temperature gradient in line 166 because “programmed temperature manner” is unclear.
-Please, give in a separate sentence the flow rate of mobile phase used for the UPLC-MS-MS analysis. I suggest giving this information in line 185, immediately after the description of the gradient of solvents.
-The authors should provide more details about the MS-MS analysis (e.g. capillary voltage, conditions at the nebulizer, range of mz considered in the study, drying conditions, number of fragmentation steps, gases feeding the MS equipment, etc. ).
-Sections 3.5.1. and 3.5.2. might be shown as a single section.
-Perhaps the authors are incorrectly using the term fixed oil, because they also analyzed some volatile compounds in it. I was wondering if you might use a better term for that. On the other hand, the authors did not discuss the data of lipids. As stated above, the authors did not justify the analysis of lipids in the introduction.
-The authors argue that the identification of compounds analyzed by UPLC-MS-MS was carried out by using libraries. However, some of these libraries only contain data for compounds analyzed by GC-MS. I do not understand how the authors compared their fragmentation patterns with those of libraries containing only data for compounds analyzed by GC-MS. MS for GC and liquid chromatography give different fragmentation patterns, which can hardly compared each to other. In my opinion this compromised the correct identification of the compounds. The authors must explain clearly how the identification of compounds was achieved. On the other hand, the discussion of these data is inexistent in the manuscript. The author might compare the phytochemical composition of the tested roots with that reported for the same species or other species of the genus Saussurea. The compositional data might also be associated with the biological effects associated with Saussurea costus.
-The authors argue that the identification of compounds analyzed by UPLC-MS-MS was also carried out by comparing their MS-MS data with those of literature, however, the authors do not indicate the specific data from literature that were used to identify the compounds of aussurea costus.
-In my opinion, the Figures 1, 2 and 3 are unnecessary and might be eliminated from the manuscript.
-There are some values marked in yellow in tables.
Reviewer 3 Report
- Saussurea costus (Falc.) Lipsch. is an accepted name
- When in what month was the material for the study of the multiannual plants were collected
- The mass spectra with electrospray ionization (ESI) in negative and positive mode range from ....... to ........ m/z. were obtained?
- What parameters of the ion source: negative and positive ion mode, gas flow rate: ….. L/min, gas temperature: …….°C, sheath gas temperature: ……°C, sheath gas flow rate: ……. L/min, nebulizer pressure: …….. psig, V Cap: ……… V, octopole RF Peak: ……. V, skimmer: …… V, and fragmentor: ……… V.
- The acquisition mode was auto MS/MS, and the collision-induced dissociation energy (CID) was …… and …….. eV, with the MS scan rate at ….. spectrum per second and …… spectra per cycle.
- Qualitative analysis of the extract from TF was performed in the auto MS/MS mode with excluded m/z at………. And ………. for the negative ion mode corresponding to the m/z of reference ions.
- Qualitative analysis of the extract from TF was performed in the auto MS/MS mode with excluded m/z at………. And ………. for the positive ion mode corresponding to the m/z of reference ions.
- The phenolic compounds in the samples were identified by comparison of the obtained MS/MS mass spectrum with literature data………..
Round 2
Reviewer 2 Report
The manuscript has been improved slighly . However, the work is still poorly justified in the introduction section. The identification of compounds is still weak. The identification of each compound should be strengthened. The compositional data might also be associated with the biological effects associated with Saussurea costus.
